# CRVR: Continuous Representation-Driven Video Frame Modulation Against rPPG Heart Rate Measurement

## Abstract

Facial video-based remote physiological measurement (rPPG) has gained prominence for its ability to non-invasively estimate vital signs such as heart rate (HR). The foundation of rPPG lies in using a camera to record facial videos at a certain frame rate, allowing the capture of rapid skin color changes necessary for HR measurement. Inspired by this property, we identified a new task, that is, to embed malicious information into facial videos by subtly modulating frames and generating frames corresponding to the modified rate. With this task, we can mislead state-of-the-art rPPG HR methods through natural and imperceptible frame modulation changes, aiming for two objectives: testing the resilience of rPPG methods against frame modulation variations and safeguarding heart rate data, which is crucial for individual privacy. However, such a task is non-trivial and should be capable of automatically adapting to different input videos and generating natural, imperceptible frame modulation perturbations along with frames corresponding to the modified rate. To address these challenges, we propose **C**ontinuous **R**epresentation-driven **V**ideo **R**esampling (CRVR), which targets precise manipulation of frame timing to subtly skew perceived HR measurements. Specifically, the CRVR method consists of two modules: Variable Frame Rate Video Resampling (VFRVR), which automatically determines the optimal resampling strategy for each frame, and Continuous Video Frame Generation (CVFG), which generates frames corresponding to the modified rate and seamlessly injects them back into the video. Extensive testing on UBFC-rPPG and PURE datasets reveals that our CRVR method successfully produces realistic, imperceptible adversarial videos that effectively mislead three different rPPG-based heart rate detection technologies.

## 1 Introduction

Remote photoplethysmography (rPPG) has been widely investigated as a prominent non-invasive technique for heart rate (HR) estimation using facial videos (Chen & McDuff, 2018; Yu et al., 2019; Liu et al., 2020; Gideon & Stent, 2021; Lu et al., 2021). The rPPG circumvents the discomfort and potential allergic reactions caused by skin-contact electrodes and wires, presenting a practical substitute for conventional contact-based devices such as electrocardiograms, thereby enhancing user comfort and accessibility (Krittanawong et al., 2021). The foundation of rPPG, as shown in Figure 1(a), lies in using a camera to record facial videos at a certain frame rate, allowing the capture of rapid skin color changes necessary for HR measurement. This process is influenced by the video's frame rate, meaning even minor adjustments can impact the accuracy of the extracted signal.

Inspired by this property, we identified a new task: embedding malicious information into facial videos by subtly altering the frame rate and generating corresponding frames for the modified rate, as shown in Figure 1(b). The goal of this task is to mislead state-of-the-art rPPG heart rate detection methods through natural, imperceptible frame modulation changes, with two primary objectives in mind: first, to evaluate the robustness of existing rPPG methods against frame modulation variations, and second, to protect heart rate data, which is a critical aspect of individual privacy. Achieving this task is particularly challenging, as it requires the system to automatically adapt to various input videos, ensuring that the alterations remain visually imperceptible while maintaining a natural flow. The solution must be capable of generating seamless frame modulation perturbations and producing

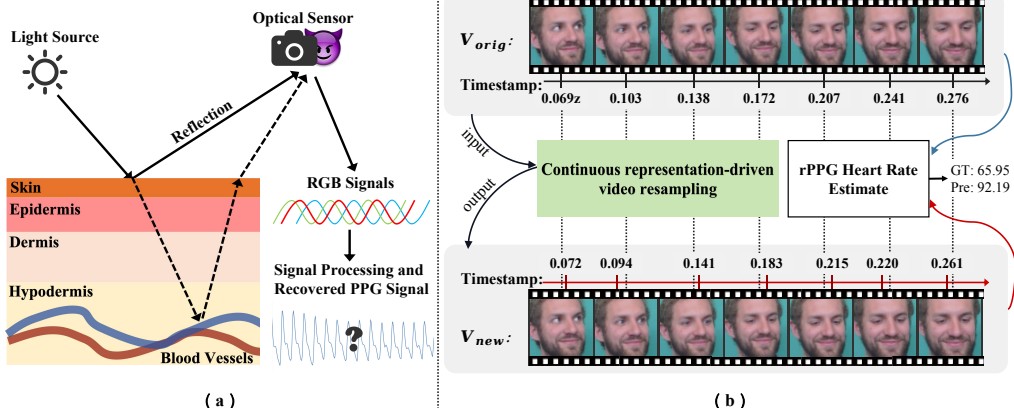

Figure 1: (a) shows the principle of rPPG for HR measurement. This method uses a camera to record facial videos and analyzes the changes in light reflected by the facial skin to extract the PPG signal. (b) shows the main idea of this work. We propose embedding malicious information into facial videos by subtly altering the frame rate and generating corresponding frames for the modified rate, achieving an effective attack while ensuring natural temporal transitions in the video.

corresponding frames that fit the modified rate, all without introducing noticeable artifacts. Such subtle modifications demand an advanced level of adaptability and precision, as the outcome should deceive even the most sophisticated rPPG models while preserving the video's authenticity from a viewer's perspective. This complexity makes our task not only technically demanding but also highly significant for testing the resilience and reliability of rPPG-based heart rate estimation systems.

To address this, we explored a range of existing attack methods and their variations. Our findings revealed that these conventional approaches often lead to significant discrepancies between consecutive frames, resulting in unnatural artifacts in the attacked videos. To develop an effective solution, we identified three primary challenges. The first challenge is *stealthiness*, which involves designing an attack strategy that subtly alters the video frame rate while preserving the visual coherence and naturalness of the video sequence. This requires not only imperceptible modifications to the frame rate but also generating frames that correspond to the modified rate, all while maintaining temporal smoothness between frames to minimize the perceptibility of the attack. In addition to stealthiness, two further challenges must be addressed to ensure the method's effectiveness. The second challenge is *generality*, which requires that the resulting HR from the attacked video significantly differs from the true HR detected by rPPG, thus protecting user privacy. The third challenge is *controllability*, meaning that the attack must enable us to manipulate the targeted HR value, either increasing or decreasing it as needed, thereby providing precise control over the outcome of the attack.

To tackle these challenges, we propose **C**ontinuous **R**epresentation-driven **V**ideo **R**esampling (CRVR), which targets precise manipulation of frame timing to subtly skew perceived HR measurements. Specifically, the CRVR method comprises two modules: Variable Frame Rate Video Resampling (VFRVR) and Continuous Video Frame Generation (CVFG). VFRVR utilizes a deep neural network to predict the optimal adversarial frame timing adjustments based on the characteristics of each frame, maintaining temporal consistency while altering the perceived frame sequences. CVFG generates frames corresponding to the modified rate seamlessly injects these adjusted frames back into the video to ensure a smooth and natural appearance.

Extensive experiments on the UBFC-rPPG and PURE datasets demonstrate the effectiveness of our CRVR method. The results show that CRVR successfully generates realistic and imperceptible adversarial videos that mislead three different rPPG-based heart rate detection technologies. The experimental results show that our method is effective in protecting user privacy, and highlight the critical vulnerabilities in rPPG HR measurement systems that rely on visual inputs.

## 2 RELATED WORKS

### 2.1 RPPG MEASUREMENT

Facial video-based remote physiological measurement (rPPG) has advanced significantly, enabling non-invasive extraction of heart rate and other vital signals from facial videos. Historically, rPPG

techniques have evolved through two major phases. Initially, conventional methods analyzed subtle color variations in specific facial regions to estimate heart rate, relying on mathematical models such as blind signal separation (Poh et al., 2010), least mean square (Li et al., 2014), majority voting (Lam & Kuno, 2015), and self-adaptive matrix completion (Tulyakov et al., 2016). These approaches were grounded in skin reflection models (Wang et al., 2016; De Haan & Jeanne, 2013) and focused on signal extraction from predetermined areas of the face. However, these traditional methods often struggled with robustness and generalizability across diverse conditions and subjects, limited by their foundational mathematical assumptions. The emergence of deep learning ushered in a transformative phase in rPPG research, leading to the development of more adaptive and powerful models. Contemporary deep learning-based approaches, such as DeepPhys (Chen & McDuff, 2018), MTTS (Liu et al., 2020), and BigSmall (Narayanswamy et al., 2024), have significantly enhanced the accuracy of heart rate detection.

## 2.2 GENERAL ADVERSARIAL ATTACKS

With the rapid development of deep learning, adversarial attacks have emerged as a significant threat to deep neural networks. To probe their vulnerabilities, researchers have developed various methods designed to deceive applications such as image classification and face recognition systems. Techniques like the Fast Gradient Sign Method (FGSM) (Goodfellow et al., 2015), iterative FGSM (Kurakin et al., 2017), and natural transformation-based attacks (Shamsabadi et al., 2020) have shown that deep learning models are susceptible to imperceptible perturbations. Furthermore, methods such as Projected Gradient Descent (PGD) (Madry et al., 2018), the Carlini-Wagner attack (Carlini & Wagner, 2017), and DeepFool (Moosavi-Dezfooli et al., 2016) have expanded the arsenal available for generating adversarial examples, proving their effectiveness across various domains. Recent advances have extended adversarial attacks to video-based systems. For instance, Universal Adversarial Perturbations (UAPs) have been adapted to target video models, misleading video classification systems through subtle, imperceptible changes between frames (Moosavi-Dezfooli et al., 2017; Papernot et al., 2017). Additionally, methods like temporal perturbations (Li et al., 2019) and spatiotemporal adversarial attacks (Liu et al., 2022) have been developed to exploit the unique dynamics of video data. Black-box attacks, which generate adversarial examples by repeatedly querying video models to construct perturbations leading to misclassification without accessing the model's internal parameters, are also evolving (Ilyas et al., 2018; Andriushchenko et al., 2020).

## 2.3 VIDEO FRAME INTERPOLATION

Video Frame Interpolation (VFI) is an established area of study within video processing, involving the synthesis of intermediate frames that do not originally exist between two consecutive video frames. This technology is utilized across various practical applications in video processing including creating slow-motion effects (Bao et al., 2019; Jiang et al., 2018), enhancing the frame rate (Bao et al., 2018; Castagno et al., 1996), compressing video (Wu et al., 2018), generating new viewpoints (Flynn et al., 2016), restoring video quality (Kim et al., 2020; Tian et al., 2020; Wang et al., 2019; Werlberger et al., 2011), and aiding in intra-prediction for video encoding (Choi & Bajić, 2019; Wu et al., 2015).

## 3 PROBLEM FORMULATION AND MOTIVATION

### 3.1 ADVERSARIAL ATTACKS AGAINST rPPG-BASED HR DETECTION

Given a facial video $\mathcal{F}$ with each frame timestamp denoted as $\mathcal{T} = \{t_i\}_{i=1}^n$, where $t_i < t_{i+1}$, and the corresponding video frames are represented by $\mathcal{F} = \{f_i\}_{i=1}^n$, the process of generating an adversarial attack video for rPPG heart rate detection can be formulated as follows:

$$\begin{aligned} \mathcal{F}' = G(\Psi(\mathcal{T}), \mathcal{F}), & \\ \text{s.t.} \quad \|f_i - f_{i+1}\| < \Delta, \quad & \forall i \in \{1, 2, \ldots, n-1\}, \\ \|t'_{i+1} - t'_i - (t_{i+1} - t_i)\| \le \xi, \quad & \forall i \in \{1, 2, \ldots, n-1\}, \end{aligned} \quad (1)$$

where $\Psi(\cdot)$ is a deep learning model that maps original timestamps $\mathcal{T}$ to attacked timestamps $\mathcal{T}' = \{t'_i\}_{i=1}^n$. Based on $\mathcal{T}'$, the generator $G(\cdot)$ creates an attacked frame sequence $\mathcal{F}' = \{f'_i\}_{i=1}^n$. The parameters $\Delta$ and $\xi$ are thresholds ensuring that generated frames maintain visual consistency and smooth temporal transitions, respectively.

Then, given an rPPG-based HR detection model $\varnothing(\cdot)$ that extracts the dominant frequency from the rPPG signal, typically using methods such as Fast Fourier Transform (FFT) (Cooley & Tukey, 1965), we design a suitable loss function, such as the cross-entropy loss for a neural network, using the HR measured from the attacked video and the original video:

$$\mathcal{L}_{att} = \arg\max(\varnothing\left(\hat{\mathcal{F}}\right), \varnothing(\mathcal{F})). \tag{2}$$

Note that during the optimization the functions $G(\cdot)$ and $\Psi(\cdot)$, the rPPG model $\varnothing(\cdot)$ is not updated. In Section 3.2, we introduce three naive implementations relevant to the adversarial video attack domain, and discuss the motivations for frame modulation attack in Section 3.3.

## 3.2 NAIVE IMPLEMENTATION

**Random Noise Addition:** Add random noise $\epsilon_t^{ran}$ to each frame $f_i$ at time $t_i$ in the video as a baseline to evaluate the overall robustness of the rPPG algorithm against non-targeted perturbations, which can be represented as,

$$\hat{\mathcal{F}}^{ran} = \{f_i + \epsilon_i^{ran}\}_{i=1}^n. \tag{3}$$

**PGD (Madry et al., 2018):** This method is an iterative enhancement version of the FGSM method, enabling more precise control over adversarial perturbations. It operates against the negative gradient of the loss function $L(\omega, f_i, y)$, where $\omega$ represents the model parameters, $f_i \in \mathcal{F}$ is the input frame, and $t_i \in [t_1, t_2, ..., t_n]$ represents the respective labels. Since the PGD algorithm is suitable for classification tasks, we let the timesteps as the label aim to make each video frame not belong to the current time step, even if each frame is different from itself. The PGD update rule for one frame is defined as:

$$f_i^{k+1} = \Pi_{f_i + \mathcal{S}}\left(f_i^k + \alpha \operatorname{sgn}(\nabla_{f_i} L(\omega, f_i, t_i))\right), \tag{4}$$

where $\mathcal{S} \subseteq \mathbb{R}^d$ denotes the permissible perturbation space, $k$ is the iteration count, sgn is the sign operator and $\alpha$ is a step size parameter reflecting a trade-off in the perturbation process.

**EMA-VFI (Zhang et al., 2023):** This method leverages inter-frame attention to explicitly extract both motion and appearance information for video frame interpolation. The inter-frame attention mechanism not only enhances appearance features by aggregating appearance information from neighboring frames but also estimates approximate motion vectors between frames. Specifically, for any patch in the current frame, its temporal neighbors are used to derive an attention map that captures their temporal correlation. The update rule for the extracted appearance feature is given by:

$$A_{f_i}^{k+1} = A_{f_i}^k + S_{f_i \to f_{i+1}} V_{f_{i+1}}, \tag{5}$$

where $S_{f_i \to f_{i+1}}$ is the attention map derived from the similarity between a patch in frame $f_i$ and its spatial neighbors in frame $f_{i+1}$, and $V_{i+1}$ represents the value derived from the appearance feature of frame $f_{i+1}$. The process simultaneously extracts motion features by estimating the displacement of each patch, providing explicit cues for the generation of intermediate frames.

## 3.3 LIMITATIONS AND MOTIVATION

**Limitations.** The previously mentioned methods, including Random Noise Addition, PGD, and EMA-VFI, have two major limitations: ❶ They lack temporal smoothness, resulting in unnatural temporal changes between video frames that may reveal signs of tampering. ❷ They introduce significant visual alterations, such as noise perturbations and interpolated frames of reduced quality, leading to unnatural visual effects. These limitations increase the risk of the rPPG system detecting the presence of an attack. Furthermore, the limited research focused on adversarial attacks in the rPPG domain underscores the importance and urgency of this study.

**Motivation.** We aim to conduct an effective adversarial frame modulation attack on facial videos for rPPG HR detection. This study is driven by three key motivations: ❶ To explore the impact of adversarial frame modulation on rPPG heart rate detection, thereby safeguarding heart rate data to protect individual privacy. ❷ To predict the optimal adversarial timing adjustment for each frame based on the characteristics of the original video, achieving an effective attack while ensuring natural temporal transitions in the video. ❸ To preserve the video's structural integrity and maintain high visual quality, ensuring that adversarial modifications are not easily perceptible to human observers.

# 4   METHODOLOGY

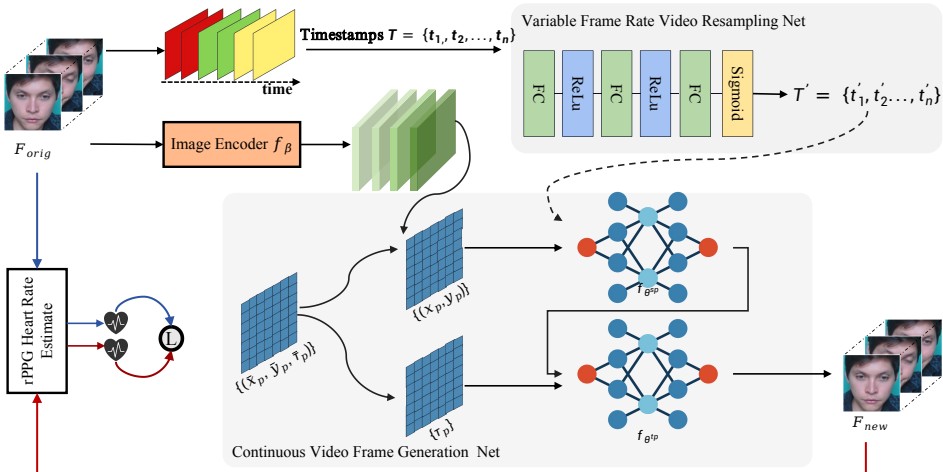

Figure 2:   The pipeline of proposed continuous representation-driven video resampling (CRVR) contains two key parts, i.e., Variable Frame Rate Video Resampling (VFRVR) and Continuous Video Frame Generation (CVFG). The original video is input into the VFRVR to predict the timestamps of the adversarial video. The CVFG then generates the corresponding frames based on the timestamps output from VFRVR and the original video features, ultimately synthesizing the adversarial video.

## 4.1   OVERVIEW

As analyzed in the previous section, achieving our goal requires predicting the optimal adversarial timing adjustment for each frame based on the characteristics of the original video, followed by generating corresponding images according to the adjusted time sequence. This approach allows for precise manipulation of the frame sequence, enabling an effective attack on rPPG facial videos. To accomplish this, we designed two modules: Variable Frame Rate Video Resampling (VFRVR) and Continuous Video Frame Generation (CVFG), as shown in Figure 2. The original video is input into the VFRVR, which predicts the time sequence of the adversarial video based on the characteristics of each frame. The CVFG then generates the corresponding frames based on the timestamps output from VFRVR, ultimately synthesizing the adversarial video with altered timestamps.

## 4.2   VARIABLE FRAME RATE VIDEO RESAMPLING MODULE

Given an original facial video $\mathcal{F} = \left\{ f_i \in \mathbb{R}^{H \times W} \right\}_{i=i}^{n}$ with n frames, and its time sequence denoted as $\mathcal{T} = \{t_i\}_{i=i}^{n}$, where $t_i < t_{i+1}$. We define a variable frame rate video resampling function $\mathcal{T}' = \Psi(\mathcal{T})$, which can be a deep neural network, to predict the adversarial timing adjustments for each frame based on the original video and its corresponding timestamps.

We aim for the generated time sequence $\mathcal{T}' = \{t_i'\}_{i=i}^{n}$ to match the input time sequence $\mathcal{T} = \{t_i\}_{i=i}^{n}$, ensuring that the generated video maintains temporal consistency with the original video. To this end, we introduce a matching loss to minimize the difference between the generated and input timestamps:

$$\mathcal{L}_{\text{match}} = \sum_{i=1}^{n} |t_i' - t_i|^2 . \tag{6}$$

In addition, to maintain temporal consistency and natural smoothness in the generated frame sequence, a temporal smoothness loss is introduced to control the differences between adjacent frames:

$$\mathcal{L}_{\text{smooth}} = \sum_{i=1}^{n-1} \left( t_{i+1}' - t_i' - (t_{i+1} - t_i) \right)^2 . \tag{7}$$

This loss ensures that changes in timestamps are not abrupt and match the relative variations in the original time sequence, thereby preserving the structure and rhythm of the video. Finally, the video

resampling function will also ensure that the length of the generated video matches the length of the original video. The CVFG module (introduced in Section 4.3) will use the newly generated adversarial timing sequence to generate the corresponding frames.

## 4.3 CONTINUOUS VIDEO FRAME GENERATION MODULE

Given the original facial video $\mathcal{F} = \left\{ f_i \in \mathbb{R}^{H \times W} \right\}_{i=1}^{n}$ and the attacked time sequence $\mathcal{T}' = \{t_i'\}_{i=i}^{n}$, consider the timestamp $t_i'$, we set a temporal window of $m$ frames whose timestamp is neighbor to $t_i'$ before and after. we aim to construct an implicit representation that captures both spatial and temporal dependencies, considering the extended temporal context of previous and future frames, and then construct the frame for timestamp $t_i'$.

Consider the image sequence $\mathcal{S} = \left\{ f_i \in \mathbb{R}^{H \times W} \right\}_{i=t_i-\frac{m}{2}}^{t_i+\frac{m}{2}}$ within the temporal window of timestamp $t_i'$, consisting of $m$ frames. We aim to construct an implicit representation $\mathcal{I}_{t_i}$ for the frame $f_{t_i}'$ corresponding to timestamp $t_i'$. This representation maps the spatial and temporal coordinates of a pixel, denoted as $p = (x_p, y_p, \tau_p)$ in the continuous domain, to its corresponding RGB value, $\mathcal{I}_{t_i}(p)$. To achieve this, we propose extending the local implicit image representation framework (Chen et al., 2021; 2024), not only focusing on frames preceding the current frame but also incorporating future frames to capture a richer temporal context. The task can be formulated as follows:

$$\mathcal{I}_{t_i}(p) = \sum_{q \in \mathcal{M}_p} \omega_q f_\theta(z_q, \text{dist}(p, q)). \tag{8}$$

The set $\mathcal{M}\mathbf{p}$ contains neighboring pixels of $\mathbf{p}$ within the temporal window of $m$ frames. The vector $\mathbf{z}\mathbf{q}$ denotes the feature of a neighboring pixel $\mathbf{q}$, and $\text{dist}(\mathbf{p}, \mathbf{q})$ measures the spatial distance between pixels $\mathbf{p}$ and $\mathbf{q}$. The function $f_\theta$ is an MLP that maps the feature of a neighboring pixel $\mathbf{q}$ to the color of pixel $\mathbf{p}$ based on their spatial distance. The final color values are obtained by aggregating the generated color values, where each value is weighted by $\omega_\mathbf{q}$. The weight $\omega_\mathbf{q}$ is determined by the volume ratio of the cube formed by $\mathbf{p}$ and $\mathbf{q}$ relative to the total neighboring volume.

Inspired by (Chen et al.), we extend our approach by dividing the implicit representation process into two phases: the first phase ensures spatial consistency, while the second expands the temporal context to incorporate both past and future frames.

**Spatial Implicit Representation:** we construct a spatial implicit representation that estimates the color value of a pixel based on its neighboring pixels across the spatial domain, for $t_i$ frame: In the first phase, we build a spatial implicit representation to estimate the color value of a pixel based on its spatial neighbors across the $t_i$ frame:

$$\mathcal{I}_{t_i}(\mathbf{p}_{t-\frac{m}{2}:t+\frac{m}{2}}) = \sum_{(x_q, y_q) \in \mathcal{M}_{(x_p, y_p)}} \omega_{(x_q, y_q)}^{sp} f_\theta^{sp}(\mathbf{z}_{(x_q, y_q)}, \text{dist}((x_p, y_p), (x_q, y_q))), \tag{9}$$

where $\mathbf{p}t - \frac{m}{2} + \frac{m}{2} = \left[ \left( x_p, y_p, t - \frac{m}{2} \right), ..., \left( x_p, y_p, t + \frac{m}{2} \right) \right]$ represents the pixel's spatial position across the entire temporal window. The function $f_{\theta^{sp}}$ is modeled as an MLP parameterized by $\theta^{sp}$, while the weight $\omega_{(x_q, y_q)}^{sp}$ is computed based on the area ratio of the rectangle formed by $(x_p, y_p)$ and $(x_q, y_q)$ compared to the neighboring areas, as done in (Chen et al., 2021).

**Temporal Implicit Representation:** After constructing the spatial representation, we expand to the temporal domain, thereby capturing temporal dependencies between frames:

$$\mathcal{I}_{t_i}(\mathbf{p}) = \sum_{\tau_q \in [t-\frac{m}{2}, t+\frac{m}{2}]} \omega_{\tau_q}^{tp} f_\theta^{tp}(\mathcal{I}_{t_i}(\mathbf{p}_{t-\frac{m}{2}:t+\frac{m}{2}}), \text{dist}(\tau_p, \tau_q)), \tag{10}$$

where $\mathcal{I}t_i(\mathbf{p}t - \frac{m}{2} + \frac{m}{2})[\tau_q]$ is the element corresponding to $\tau_q$ in $\mathcal{I}t_i(\mathbf{p}t - \frac{m}{2} + \frac{m}{2})$, and $f_{\theta^{tp}}(\cdot)$ is an MLP used to predict the color value of pixel $\mathbf{p}$ from the feature vector $\mathcal{I}t_i(\mathbf{p}t - \frac{m}{2} + \frac{m}{2})[\tau_q]$. The weight $\omega_{\tau_q}^{tp}$ is based on the temporal distance between the frame being constructed and its neighboring frames. We can then simplify the formulation as follows:

$$\mathcal{I}_{t_i}(\mathbf{p}) = \text{CVFG}(\mathbf{p}, \mathcal{S} \mid f_\beta, f_\theta^{sp}, f_\theta^{tp}), \tag{11}$$

where $f_\beta$ serves as an encoder network that extracts features from the pixels.

By iterating through all discrete coordinates within the frame, we can reconstruct the frame at timestamp $t_i'$ as $f_{t_i}'$. All the frames corresponding to the time sequence after the attack are concatenated to form the adversarial video, $\mathcal{F}' = \bigoplus_{i=1}^{n}\{\mathcal{F}\}_{i=1}^{n}$, where $\bigoplus$ denotes frame concatenation operate. We use the reconstruction loss to directly compare the pixel-level differences between the generated frames and the original frames to pre-train this module:

$$\mathcal{L}_{\text{reconstruction}} = \sum_{i=1}^{n} \|f_{t_i}' - f_{t_i}\|_1. \tag{12}$$

### 4.4 IMPLEMENTATION DETAILS

**Training process.** We first pre-train the CVFG module and then fine-tune this module. During pre-training, the timestamps of the original video is input, and the generated frames at the corresponding timestamps of the original video are used to calculate the loss according to Equation 12 for training. To avoid unnatural jumps or ghosting effects in the generated video, we employ a motion constraint loss to ensure that the motion information between adjacent frames remains consistent:

$$\mathcal{L}_{\text{motion}} = \sum_{i=1}^{n-1} \|(f_{t_{i+1}}' - f_{t_i}') - (f_{t_{i+1}} - f_{t_i})\|^2. \tag{13}$$

The loss term measures the changes in differences between adjacent frames to maintain consistency of the dynamic information in the generated video with that of the original video, ensuring that the motion effect during video playback appears natural.

**Loss function.** Our goal is to generate videos that achieve an effective attack, so we propose an attack effectiveness loss $\mathcal{L}_{\text{attack}}$ using the attacked video and the original video:

$$\mathcal{L}_{\text{attack}} = |\phi(F') - \phi(F)|, \tag{14}$$

where $\phi(\cdot)$ represents the heart rate measurement method. Combining all the above loss terms, we obtain the total loss function $\mathcal{L}_{\text{total}}$ as follows:

$$\mathcal{L}_{\text{total}} = \mathcal{L}_{\text{attack}} + \lambda_1(\mathcal{L}_{\text{match}} + \mathcal{L}_{\text{smooth}}) + \lambda_2\mathcal{L}_{\text{motion}}. \tag{15}$$

Since $\mathcal{L}_{\text{match}}$ and $\mathcal{L}_{\text{smooth}}$ are only related to the VFRVR module, and $\mathcal{L}_{\text{motion}}$ is only related to the CVFG module, we introduce only two parameters, $\lambda_1$ and $\lambda_2$, for these modules.

## 5 EXPERIMENTAL EVALUATION

In this section, we conduct a series of experiments to evaluate CRAD's attack efficacy under various previously discussed settings. It should be emphasized that the results for each scenario are averaged across three trials to minimize the impact of different random seeds. In line with the design specifications of rPPG HR measurement, we have configured the reconstruction range to focus on the facial region rather than the entire image, significantly reducing time costs.

**Datasets.** We conduct our experiments on two public datasets: UBFC-rPPG (Bobbia et al., 2019), PURE (Stricker et al., 2014). UBFC-rPPG consists of 42 facial videos with simultaneously recorded PPG signals and heart rates. We follow (Lee et al., 2020) to discard subjects of indices 11, 18, 20 and 24 because their heart rates were inappropriately recorded. PURE contains 60 facial videos from 10 subjects. During the data collection process, subjects were asked to perform six kinds of head motions (small rotation, medium rotation, slow translation, fast translation, talking and steady) in front of the camera for one minute. We follow (Gideon & Stent, 2021) to discard the first two samples because their PPG waveforms were strongly corrupted.

**Metrics.** We employ Mean Absolute Error (MAE), Root Mean Square Error (RMSE), and Pearson Correlation (r) as metrics to evaluate HR accuracy, adhering to the methodology used in (Song et al., 2021; Gideon & Stent, 2021). These metrics assess the discrepancy between predicted HR and actual measurements, thus gauging the precision of HR estimates. MAE is computed as the average of the absolute differences between predicted HR and ground truth, with lower values indicating greater accuracy. RMSE is defined as the square root of the mean of the squared discrepancies between predicted and actual HR values, where lower scores reflect superior accuracy. Pearson Correlation (r) quantifies the strength of the linear relationship between predicted HR values and the ground truth,

Table 1: Performance of multiple rPPG HR methods when subjected to different attack approaches. The results are reported on UBFC-rPPG and PURE datasets.

| rPPG Model | Attacks | UBFC | | | PURE | | |
|---|---|---|---|---|---|---|---|
| | | MAE↓ | RMSE↓ | $r$ ↑ | MAE↓ | RMSE↓ | $r$ ↑ |
| RemotePPG | wo.Atk | 3.620 | 4.983 | 0.959 | 2.348 | 3.016 | 0.991 |
| | Random noise | 4.420 | 5.776 | 0.902 | 2.877 | 3.745 | 1.000 |
| | PGD | 21.675 | 29.664 | 0.426 | 20.589 | 27.983 | 0.432 |
| | Square Attack | 30.192 | 34.954 | 0.403 | 25.492 | 31.545 | 0.418 |
| | EMA-VFI | 4.150 | 5.201 | 0.910 | 6.782 | 9.021 | 0.703 |
| | Ours | 36.790 | 38.710 | 0.305 | 35.061 | 37.150 | 0.374 |
| Physformer | wo.Atk | 5.504 | 10.879 | 0.623 | 4.543 | 10.376 | 0.694 |
| | Random noise | 7.585 | 10.263 | 0.607 | 6.872 | 9.781 | 0.659 |
| | PGD | 27.108 | 35.914 | 0.327 | 25.743 | 34.289 | 0.302 |
| | Square Attack | 36.998 | 38.730 | 0.249 | 31.473 | 36.507 | 0.276 |
| | EMA-VFI | 6.900 | 9.410 | 0.613 | 7.203 | 12.074 | 0.561 |
| | Ours | 37.605 | 39.480 | 0.241 | 36.430 | 38.091 | 0.283 |
| BigSmall | wo.Atk | 1.048 | 2.585 | 0.986 | 1.908 | 6.549 | 0.931 |
| | Random noise | 2.170 | 3.638 | 0.962 | 2.760 | 5.184 | 0.905 |
| | PGD | 25.460 | 26.874 | 0.496 | 24.379 | 25.987 | 0.471 |
| | Square Attack | 30.006 | 37.890 | 0.309 | 27.006 | 31.987 | 0.390 |
| | EMA-VFI | 2.092 | 3.240 | 0.972 | 4.810 | 6.032 | 0.880 |
| | Ours | 35.561 | 37.142 | 0.310 | 33.049 | 35.210 | 0.302 |

with values nearing 1 denoting a strong positive correlation.

**Baseline.** We have selected four baseline methods for comparison with our CRAD method, chosen for their relevance to the rPPG domain and their capacity to offer a thorough evaluation of adversarial techniques. These include Random Noise, PGD (Madry et al., 2018), and EMA-VFI (Zhang et al., 2023). Additionally, we incorporate Square Attack (Andriushchenko et al., 2020), a black-box adversarial method that subtly alters a minimal number of pixels to amplify the loss in the target model. This inclusion serves to underscore the unique challenges of light manipulation versus more localized pixel-level disturbances.

**rPPG models.** We assess the proposed attacks and baseline methods on multiple facial video-based remote physiological measurement models, including PhyNet (Yu et al., 2019), MTTS (Liu et al., 2020), RemotePPG (Gideon & Stent, 2021), EfficientPhys (Liu et al., 2021), Physformer (Yu et al., 2022), and BigSmall (Narayanswamy et al., 2024). To ensure optimal performance, all these models require pre-processing of face images using MTCNN (Zhang et al., 2016), which we apply consistently throughout the evaluation.

## 5.1 COMPARATIVE ANALYSIS OF ADVERSARIAL ATTACKS ON RPPG HR MEASUREMENT

Our evaluation extensively investigated the resilience of different rPPG heart rate detection models against a variety of adversarial attacks, as illustrated in Table 1. The results are reported on both UBFC-rPPG and PURE datasets. This analysis focuses on understanding how adversarial attacks disrupt the accuracy of heart rate measurements and assesses the robustness of the rPPG models to such manipulations. The data reveals substantial variability in how different models withstand similar adversarial conditions. Notably, PGD and Square Attack significantly deteriorate the performance of the RemotePPG model, with MAE and RMSE values drastically increasing. This highlights the model's vulnerability to gradient-based and pattern disruption attacks. EMA-VFI is a frame insertion method. Although it is technically feasible, it can usually only insert the generated frames between two frames. This method changes the frame rate of the video and is visually perceptible. Because it does not take into account the natural coherence of the video. Therefore, if you want to ensure the frame rate as a prerequisite, you can only regenerate the intermediate frames by the previous and next frames to conduct attack tests. In this case, the heart rate will hardly change. In practical applications, the adversarial effect of this method is limited because it is not enough to achieve continuous misleading of heart rate measurement without attracting attention. Our method consistently leads to the highest MAE and RMSE in almost all cases, demonstrating its effectiveness

in corrupting heart rate detection. Additional results regarding the ablation study and the effectiveness analysis can be found in the Appendix.

## 5.2 COMPARATIVE VISUAL ANALYSIS OF FRAME INTERPOLATION METHODS

Figure 3 illustrates a series of video frames subjected to different adversarial attacks, including PGD, EMA-VFI, and our method, along with their impact on the derived rPPG signals. The original frames are outlined in green boxes to serve as a reference point, contrasting sharply with the red boxes which mark the adversarially altered frames. The comparison clearly illustrates how conventional frame interpolation techniques, such as PCA Attack and EVM-WFT, generally insert intermediate frames between existing ones, which limits their ability to produce frames at arbitrary timestamps. CRAD

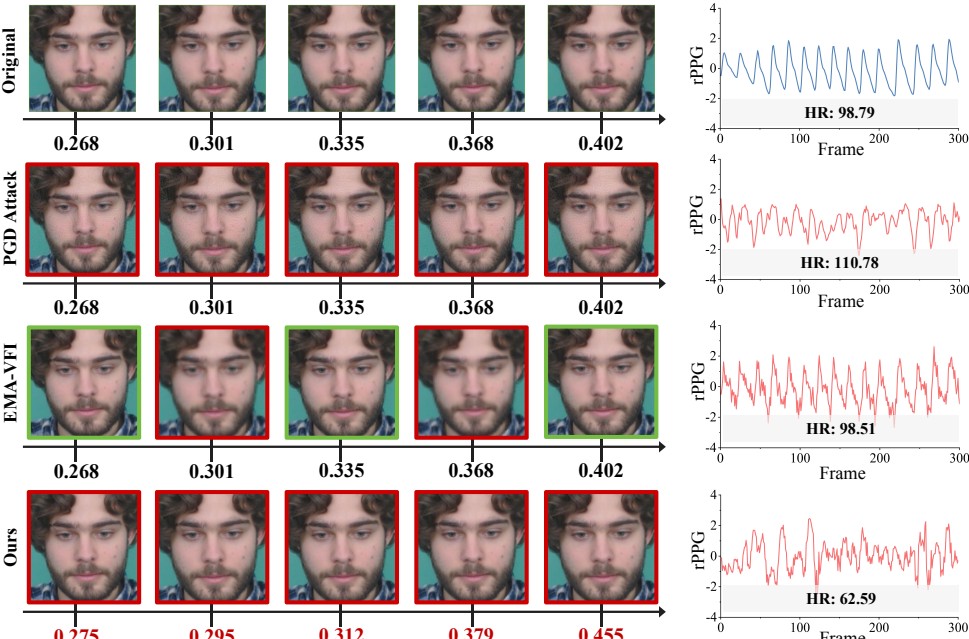

Figure 3: Visual and quantitative analysis of adversarial attacks on rPPG HR measurement. Green boxes highlight original frames, while red boxes and red text indicate frames and timestamps affected by adversarial attacks. The right side displays the corresponding rPPG signal.

stands out for its ability to seamlessly generate frames at any given moment within the video timeline, unlike traditional methods that are restricted to creating frames only at predetermined intervals. This capability allows CRAD to subtly alter perceived heart rates by crafting frame sequences that introduce temporal perturbations indiscernible to the naked eye yet effective enough to mislead advanced rPPG detection systems.

## 6 CONCLUSIONS

In this study, we presented a novel adversarial approach to facial video-based remote physiological measurement (rPPG) heart rate estimation through the Continuous Representation-driven Video Resampling (CRVR) method. This method effectively embeds malicious information into facial videos by subtly modulating frames, producing natural and imperceptible perturbations that mislead state-of-the-art rPPG heart rate methods. Extensive evaluations on the UBFC-rPPG and PURE datasets demonstrate that CRVR can consistently generate adversarial videos that appear realistic while misleading multiple rPPG-based heart rate detection algorithms. This work reveals critical vulnerabilities in current rPPG systems, highlighting the need for enhanced robustness against adversarial attacks to protect privacy and ensure accurate physiological assessments. Future research should focus on developing effective countermeasures and exploring practical applications to strengthen the security and reliability of remote physiological monitoring technologies.

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
