# OpenReview forum: "CRVR: Continuous Representation-Driven Video Frame Modulation Against rPPG Heart Rate Measurement"
_ICLR.cc/2025/Conference — ICLR 2025 Conference Withdrawn Submission_

### Official Review · Reviewer_UWXN · 2024-10-28

**Soundness:** 2
**Presentation:** 2
**Contribution:** 2
**Rating:** 3
**Confidence:** 5

**Summary:**

The paper introduces an adversarial attack technique targeting remote photoplethysmography (rPPG) systems. The proposed approach, Continuous Representation-driven Video Resampling (CRVR), aims to disrupt HR measurements by subtly modulating video frames to deceive state-of-the-art rPPG-based systems. This is done while keeping the modifications imperceptible to the viewer, thus protecting individual privacy and testing the robustness of these models. The CRVR framework comprises two main modules: 1. Variable Frame Rate Video Resampling (VFRVR) adjusts frame timings to manipulate the temporal structure of the video. 2. Continuous Video Frame Generation (CVFG) generates new frames based on the modified timestamps.

**Strengths:**

1. The paper well presents the motivation and limitations of previous adversarial methods.
2. The paper first considers the digital adversarial attacks in the rPPG measurement task.

**Weaknesses:**

1. The method part is not well presented.
    - Sec. 4.3 Continuous Video Frame Generation Module has many typos in formulas and symbols.
    - The main figure (fig.2) is not well aligned with the main text. The loss function is not mentioned in the figure. Some symbols in the main text is not marked in the figure.
    - The paper proposes using new time stamps to resample the video. However, the intuition and motivation to use new time stamps are not clear.
    - In the loss function, eq. 14 does not make sense. Why should the target heart rate and the original heart rate be similar?
2. The experiment section also lacks some quantitative results and important baselines.
    - There are no quantitative results about the visual quality of new videos as the authors emphasize the visual imperceptible in the introduction.
    - Previous works[1,2,3] have already proposed some rPPG editing methods, so these works should also be compared in the experimental part.
    - Authors mention several rPPG models, but only three are used in the results table 1.
    - If the rPPG measurement model during training is different from the model during the testing stage, how about the performance?



[1] Chen M, Liao X, Wu M. PulseEdit: Editing physiological signals in facial videos for privacy protection[J]. IEEE Transactions on Information Forensics and Security, 2022, 17: 457-471.

[2] Sun Z, Li X. Privacy-phys: Facial video-based physiological modification for privacy protection[J]. IEEE Signal Processing Letters, 2022, 29: 1507-1511.

[3] Hsieh C J, Chung W H, Hsu C T. Augmentation of rPPG benchmark datasets: Learning to remove and embed rPPG signals via double cycle consistent learning from unpaired facial videos[C]//European Conference on Computer Vision. Cham: Springer Nature Switzerland, 2022: 372-387.

**Questions:**

Please see the weakness part.

---

### Official Review · Reviewer_36GK · 2024-10-29

**Soundness:** 3
**Presentation:** 3
**Contribution:** 2
**Rating:** 5
**Confidence:** 5

**Summary:**

Remote physiological measurement (rPPG) based on facial video has been valued for its ability to noninvasiveness assess vital signs such as heart rate (HR). rPPG technology uses a camera to record video of a face at a certain frame rate, which captures rapid changes in skin tone and then measures heart rate. Inspired by this feature, the paper proposes a new task: embedding malicious information into face videos to mislead rPPG heart rate detection by subtly adjusting video frames. The goal of this task is to test rPPG's ability to resist frame adjustments while protecting heart rate data and ensuring personal privacy.

To achieve this goal, this paper develops a method called Continuous representation-driven video resampling (CRVR). CRVR consists of two parts: the first part (VFRVR) automatically determines the best way to adjust each frame, and the second part (CVFG) generates new frames that match the adjusted frame rate. Testing results on two datasets showed that the CRVR approach successfully generated natural and hard-to-detect antagonistic videos, effectively misleading multiple rPPG heart rate detection techniques. This demonstrates the effectiveness of the approach in protecting privacy and evaluating the robustness of rPPG technology.

**Strengths:**

The thesis statement is clear, and the thesis question definition is very clear.
This paper gives the principle of rPPG, which is convenient for layman to understand the whole paper.

**Weaknesses:**

1. The design of each method does not give enough motivation. The description of the entire framework only gives the how to do it, without explaining the specific reasons for each step.
2. Innovation is not highlighted. CRVR and CVFG are very general design ideas.
3. Experimental comparison is very insufficient. They compared the methods of video as input, and did not compare the methods of manual design features as input. For example, RhythmNet: End-to-end Heart Rate Estimation from Face via Spatial-temporal Representation
4. The data sets that the methods compare are very simple. The frame rate of these data sets is fixed, and the head motion is weak. It is recommended to use larger or more challenging data sets for evaluation.

**Questions:**

Please refer mainly to the weak points section, I think it is difficult for me to improve my score through a rebuttal. The innovative, systematic and in-depth nature of this paper is below than ICLR.

---

### Official Review · Reviewer_QPUt · 2024-11-03

**Soundness:** 2
**Presentation:** 2
**Contribution:** 2
**Rating:** 3
**Confidence:** 5

**Summary:**

The presented work proposes a method for modulating frames representing human faces in order to modify physiological measurements extracted from such videos.  The introduced approach consists of two phases: Representation-driven Video Resampling aimed at manipulating frame timing and Continuous Video Frame Generation, used to generate frames and inserts them back to the video. The method was tested on two benchmarks showing that it can mislead heart rate detection algorithms.

**Strengths:**

1. The idea of frame generation combined with resampling to manipulate facial representations and thus modify the physiological signals is interesting. Even though it's not novel, the results show its effectiveness.
2. The method is verified using some of the quite recent algorithms for HR estimation

**Weaknesses:**

1. Explanation of equations should be revisited to improve clarity, especially equation 9,10,11 - for example please explain "is the element corresponding to τq in Iti" in more detail what do you mean by element. Equation 9 "where pt −m/2 +m/2" - is t missing here? Same in Equation 10' description.
2. The work should include ablation study for all introduced parameters to provide more in-depth evaluation of the method and justification of its robustness.
3. It is not clear how the split for train/val/test was done among subjects. Did you ensure there is no person overlap between the sets?
4. Table 1 should contain std. Arrows next to MAE, RMSE, r should be in opposite directions in your case, as you are looking for bigger errors values and lower correlation. Please highlight the best results in the table.
5. Limitation of other methods is that they change the frame rate. Doesn't your method change it as well? Please explain in more detail as it's not clear. If it changes the frame rate, then wouldn't simple detection of frame rate identify invalid videos?
6. Figure 3 left part doesn't contribute much, as differences are not visible. Please either highlight differences or skip facial representations. Besides that it would be interesting to see how the entire final video looks when it comes to smoothness and whether the inserted frames are indeed not visible to the naked eye.

**Questions:**

1. Since many of HR estimation methods are designed to work in real time scenarios, it would be interesting to explain how the proposed method can be introduced in real time.
2. Page 7,8,9 - what is CRAD? Please ensure to explain all abbreviations

---

### Official Review · Reviewer_MYoJ · 2024-11-03

**Soundness:** 2
**Presentation:** 2
**Contribution:** 3
**Rating:** 3
**Confidence:** 3

**Summary:**

This paper aims to add malicious information into facial videos making them inappropriate to extract accurate heart rate signal, thereby preventing privacy leakage. Additionally, the paper aims to evaluate the resilience of the existing remote photoplethysmogram (rPPG) methods in extracting the heart rate signal from tampered videos. To do so, the authors propose **Continuous
Representation-driven Video Resampling (CRVR)** comprising two modules, **Variable Frame Rate Video Resampling
(VFRVR)**, that determines the optimal resampling strategy for each frame of the facial video, and **Continuous Video Frame Generation (CVFG)**, that generates new frames corresponding to the modified rate of the VFRVR module and then injects them into the video. The authors test their method with two datasets, and three rPPG methods and show that their attack method leads to heavy degradation in the predicted heart rate of the attacked videos.

**Strengths:**

The paper introduces a novel method which aims at addressing three important challenges with regards to manipulating the facial video content for incorrect rPPG estimations:

- Stealthiness: Subtly alter the video frame rate to preserve visual coherence.
- Generality: The heart rate from the original and malicious video should be significantly different.
- Controllability: The user should have the ability to manipulate the targeted heart rate.

These are very relevant and appropriately identified challenges. Remote PPG is an interesting technology, as it only requires the video of a face to extract something as intricate as a person's vitals including heart rate, blood oxygen level, heart rate variation, and others. Although, it already works with a face which is an important and ubiqutuos biometric, the heart rate signal in itself is also used as a biometric which is a serious cause of concern.

The authors compare their method with 4 different attack methods, and 3 different rPPG methods.

**Weaknesses:**

I believe the paper has the following weaknesses:

**Technical:**
- The explanation of the VFRVR, i.e. the video frame rate resmapling is unclear to me. On one hand we want the extracted heart rate of the malicious video to be different from the original video (Line 085 - *generality* challenge), but at the same time, the use of L_match (Line 267), L_attack (Line 349), aims to minimize the difference between the tampered and the original timestamps and the heart rate. This seems counter-intuitive and the authors should provide further explanation as to how these loss functions supplement the training procedure while achieving the required output of a visually coherent but tampered heart rate measurement.

- On the same lines as the previous point, the heart rate values need to be present in a certain range, say 45-200 bpm. The VFRVR should typically have some fidelity to map the malicious video's HR to a certain specified range of heart rate (Line 086 - *controllability* challenge) which seems to be missing. The authors should demonstrate through experiments or analysis as to how does their method allow for targetting specific heart rate ranges say, a low range (45-60 bpm), or high range (150-180 bpm), or any other range.

- Evaluation details of the rPPG methods seems to be missing. Surely, the author's method gives vastly degraded rPPG metrics, but how were the rPPG methods trained on more importantly on what videos? Were it the untampered videos of the same dataset? What was the training routine and dataset splits followed? The authors should introduce a dedicated subsection covering the training procedure, hyperparameter settings, data pre-processing, splitting, etc.

- On the same lines, more challenging datasets such as COHFACE, PURE (compressed - HRCNN - Spetlik et al. BMVC 2018), others need to be evaluated by the authors to check for the robustness of their method.

- In the subsection starting at Line 409, the authors mention that they use 6 rPPG methods for the comparison, whereas in Table 1, and also in the abstract, they mention use of only 3 rPPG methods. The authors should revise the claim or provide further results to substaniate this claim.

- While visual coherence of the malicious videos is a centerpoint of the authors' work (Line 080 - *stealthiness* challenge), there is no objective/numerical evidence to support the claim in the results. The authors should compare the video quality using metrics such as PSNR, SSIM, and others.

**Writing:**
- The paper needs to be thoroughly proofread to remove any gramatical/factual errors to enhance the reading experience.

- The first line of the abstract itself, Line 13, mentions an incorrect abbreviation. The authors abbreviated 'remote physiological measurement' as 'rPPG', whereas the correct abbreviation would be RPM. rPPG stands for remote photoplethysmography which I believe the authors meant to use in the paper, Line 038 onwards.

- Line 278, incorrect grammar, '.. after. *w*e ...'.

- In lines 292, 308, 318, 319, the subscripts in the text need to be fixed to provide for correct indexing and also to replicate the notation used in the preceding equations.

- From Line 359 onwards, there are mulitple mentions of 'CRAD', which I think is a typo for 'CRVR' or 'CRVR' Attack? 'CRAD' needs to be properly defined before usage in the text.

- In Table 1, needs proper highlighting of the results for enhanced readability.

- In Line 410, the authors made a typo referring to the method by Yu et al. 2019, as 'PhyNet' whereas it should be 'PhysNet'.

- In lines 427-428, the authors use the pronoun 'you' to address to the reader, which I believe to be unfit in the academic setting. Instead the authors should use more formal and professional language. For example, 'Therefore, if *one* wants to ensure the ....'.

- In Figure 3, the red text should be present for the timestamps of the affected frames for all the methods and not just the authors' method at the bottom row to maintain consistency.

**Questions:**

For questions, kindly address the **Technical** subsection of the Weaknesses.

For suggestions, kindly incorporate the **Writing** subsection of the Weaknesses and if required then some points that will improve/ throw further light on some points of the **Technical** subsection of the Weaknesses.

---

### Note · Authors · 2024-11-12

**Comment:**

We would like to express our gratitude to the reviewers for their valuable feedback. After careful consideration of the comments and the improvements suggested, we have decided to withdraw our submission to allow sufficient time to address these insights thoroughly. Thank you for the opportunity and for the reviewers' constructive input, which will be invaluable in refining our work.

**Withdrawal Confirmation:**

I have read and agree with the venue's withdrawal policy on behalf of myself and my co-authors.